# Low Concentration of the Neutrophil Proteases Cathepsin G, Cathepsin B, Proteinase-3 and Metalloproteinase-9 Induce Biofilm Formation in Non-Biofilm-Forming *Staphylococcus epidermidis* Isolates

**DOI:** 10.3390/ijms23094992

**Published:** 2022-04-30

**Authors:** Itzia S. Gómez-Alonso, Sergio Martínez-García, Gabriel Betanzos-Cabrera, Esmeralda Juárez, María C. Sarabia-León, María Teresa Herrera, Fernando Gómez-Chávez, Luvia Sanchez-Torres, Sandra Rodríguez-Martínez, Mario E. Cancino-Diaz, Jorge Cancino, Juan C. Cancino-Diaz

**Affiliations:** 1Laboratorio de Inmunomicrobiología, Departamento de Microbiología, Escuela Nacional de Ciencias Biológicas, Instituto Politécnico Nacional, Mexico City 11340, Mexico; itziasidney@gmail.com (I.S.G.-A.); sergiomargar@hotmail.com (S.M.-G.); 2Área Académica de Nutrición, Instituto de Ciencias de la Salud, Universidad Autónoma del Estado de Hidalgo, Pachuca 42160, Mexico; gbetanzo@uaeh.edu.mx; 3Departamento de Investigación en Microbiología, Instituto Nacional de Enfermedades Respiratorias “Ismael Cosió Villegas”, Mexico City 14080, Mexico; ejuarez@iner.gob.mx (E.J.); carmen.sarabia@iner.gob.mx (M.C.S.-L.); teresa_herrera@iner.gob.mx (M.T.H.); 4Laboratorio de Enfermedades Osteoarticulares e Inmunológicas, Sección de Estudios de Posgrado e Investigación, Escuela Nacional de Medicina y Homeopatía, Instituto Politécnico Nacional, Mexico City 073240, Mexico; fgomezch@ipn.mx; 5Laboratorio de Inmunidad Innata, Departamento de Inmunología, Escuela Nacional de Ciencias Biológicas, Instituto Politécnico Nacional, Mexico City 11340, Mexico; luviasanchez@hotmail.com (L.S.-T.); sandrarodm@yahoo.com.mx (S.R.-M.); mecancinod@gmail.com (M.E.C.-D.); 6Programa Moscafrut SADER-IICA, Metapa de Domínguez 30860, Mexico; jorge.cancino.i@senasica.gob.mx

**Keywords:** *Staphylococcus epidermidis*, biofilm, proteinase-3, neutrophil, commensal, cathepsin G, cathepsin B, metalloproteinase-9

## Abstract

Neutrophils play a crucial role in eliminating bacteria that invade the human body; however, cathepsin G can induce biofilm formation in a non-biofilm-forming *Staphylococcus epidermidis* 1457 strain, suggesting that neutrophil proteases may be involved in biofilm formation. Cathepsin G, cathepsin B, proteinase-3, and metalloproteinase-9 (MMP-9) from neutrophils were tested on the biofilm induction in commensal (skin isolated) and clinical non-biofilm-forming *S. epidermidis* isolates. From 81 isolates, 53 (74%) were *aap*^+^, *icaA*^−^, *icaD*^−^ genotype, and without the capacity of biofilm formation under conditions of 1% glucose, 4% ethanol or 4% NaCl, but these 53 non-biofilm-forming isolates induced biofilm by the use of different neutrophil proteases. Of these, 62.3% induced biofilm with proteinase-3, 15% with cathepsin G, 10% with cathepsin B and 5% with MMP -9, where most of the protease-induced biofilm isolates were commensal strains (skin). In the biofilm formation kinetics analysis, the addition of phenylmethylsulfonyl fluoride (PMSF; a proteinase-3 inhibitor) showed that proteinase-3 participates in the cell aggregation stage of biofilm formation. A biofilm induced with proteinase-3 and DNAse-treated significantly reduced biofilm formation at an early time (initial adhesion stage of biofilm formation) compared to untreated proteinase-3-induced biofilm (*p* < 0.05). A catheter inoculated with a commensal (skin) non-biofilm-forming *S. epidermidis* isolate treated with proteinase-3 and another one without the enzyme were inserted into the back of a mouse. After 7 days of incubation period, the catheters were recovered and the number of grown bacteria was quantified, finding a higher amount of adhered proteinase-3-treated bacteria in the catheter than non-proteinase-3-treated bacteria (*p* < 0.05). Commensal non-biofilm-forming *S. epidermidis* in the presence of neutrophil cells significantly induced the biofilm formation when multiplicity of infection (MOI) 1:0.01 (neutrophil:bacteria) was used, but the addition of a cocktail of protease inhibitors impeded biofilm formation. A neutrophil:bacteria assay did not induce neutrophil extracellular traps (NETs). Our results suggest that neutrophils, in the presence of commensal non-biofilm-forming *S. epidermidis,* do not generate NETs formation. The effect of neutrophils is the production of proteases, and proteinase-3 releases bacterial DNA at the initial adhesion, favoring cell aggregation and subsequently leading to biofilm formation.

## 1. Introduction

Biofilms are bacterial communities adhered to biotic or abiotic surfaces, where bacteria are embedded inside extracellular polymeric substances produced by the same bacteria and is considered a virulence factor since the biofilm protects bacteria from the host’s immune system and antibiotics [1]. Biofilm-forming *Staphylococcus epidermidis* isolates are common in cases of foreign body infections such as medical devices [2]. Biofilm formation by these isolates explains their pathogenicity mechanism since the biofilm is a protective factor against antibiotics and the host’s immune response [1]. However, there are cases of infective isolates of *S. epidermidis* with a non-biofilm-forming phenotype (in-vitro) with a frequency of isolation of around 30%–50% [3]. In these cases, it is difficult to explain the pathogenicity mechanism since the main *S. epidermidis’* virulence factor is the biofilm.

Biofilm formation is a complex process that involves different molecules. The development of biofilm includes different steps: initial adhesion, cell aggregation, and biofilm breakdown [4]. The initial adhesion involves molecules of bacterial surface that are dependent on the support material; for the case of abiotic surfaces, the autolysin E (AtlE) protein [5] and teichoic acid are involved; for biotic surfaces, the SdrG and SdrF proteins [6] are the participants. Concerning the cell aggregation, two mechanisms are known; one involves the *icaADBC* operon, and the other is related to the accumulation-associated protein (Aap) and the extracellular matrix binding protein (Empb) proteins. The *icaADBC* operon synthesizes the poly-N-acetyl glucosamine (PNAG or PIA) polymer [7] and is regulated by the IcaR protein [8]. The expression of the *icaADBC* operon is complex since many factors are involved in its expression; some of them are NaCl [9], glucose [10], temperature [11], and ethanol [9], which can enhance or turn off the biofilm formation. The *icaADBC* operon is the most commonly found in biofilm-forming isolates [12]. The other mechanism is the protein-dependent biofilm that generates a biofilm formed mainly by the Aap [13,14] or Empb proteins [15,16]. Isolates that produce biofilm in this way have the *icaADBC* operon mutated and lack the PNAG/PIA [14,17]. Additionally, the release of chromosomal DNA (extracellular DNA; eDNA) by lysed bacteria help as a support for the biofilm formation, and *S. epidermidis* AtlE protein is involved in this process [18]. Regarding the last step of biofilm formation, there is little information on how this process occurs, but phenol soluble modulins are involved in the biofilm breakdown [4]. There is extensive information on biofilm-forming isolates; however, for the case of non-biofilm-forming isolates, it is unknown whether these isolates can form biofilm under certain conditions or remain unable to do this process [19].

The Aap protein is involved in protein-dependent biofilm formation. Rohde et al. (2005) demonstrated that the Aap protein is proteolytically processed between its A and B domains (internal domain) to expose the B domain [20]. Two B domains interact with each other to bind the bacteria and produce cellular aggregation in biofilm formation [20]. The proteolysis of Aap can be done by *S. epidermidis*’ own proteases such as metalloprotease A (SepA) [21], or by external proteases such as trypsin, elastase and cathepsin G [20], the last two produced by neutrophils. Biofilm formation was induced in a non-biofilm-forming clinical *S. epidermidis* 1457 strain treated with elastase or cathepsin G, indicating that neutrophil proteases could be involved in the induction of biofilm formation [20]. Another molecule that needs proteolytic processing to carry out its function is the AtlE protein; its role is bifunctional; on one hand, it participates in the bacterial lysis for the death process (release of eDNA), and on the other hand, it has a role in the adhesion to abiotic surfaces [18,22], suggesting that the SepA protease from *S. epidermidis* is the one involved in this proteolytic process to AtlE [23] and it is known that a component of bacterial biofilm is the extracellular DNA (eDNA) [23].

Neutrophils are the first cells recruited at the infection site, and when they are activated by bacterial antigens, they perform bacterial phagocytosis. In the process, the granular content is released into the phagolysosome, which contains different types of enzymes, among them several proteases [24]. Nevertheless, their bactericidal activity does not always occur, as known with biofilm-forming *Pseudomonas aeruginosa* that, in the presence of neutrophils, increases biofilm formation. The mechanism to increase biofilm is by the releasing of DNA and F-actin from the neutrophil, a mechanism called neutrophils extracellular traps (NETs) [25,26]. NETs are an immune response by neutrophils, consisting of the release into the extracellular space of a DNA mesh that encloses histones and antimicrobial proteins to immobilize microorganisms and kill them.

*S. epidermidis’* biofilm interferes with the immune activity of macrophages and neutrophils. *S. epidermidis’* biofilm is protective from phagocytosis and restricts the production of pro-inflammatory cytokines by macrophages, regardless of the morphotype (PNAG/PIA, Embp-, or Aap-biofilms) [16,27,28] and avoiding the immune response against *S. epidermidis*. The caspase-1 activation [29] and, therefore, the release of active IL-1β [30] produced by neutrophils is reduced by the *S. epidermidis* biofilm. It has been observed that GroEL protein within the biofilm and PNAG/PIA can activate neutrophils by releasing antimicrobial peptides [31], the degranulation of lactoferrin and elastase [32], and by the release of DNA and the inflammatory cytokine MRP-14 [33]. Besides, the PNAG/PIA macromolecule of biofilm can generate faster neutrophils recruitment and bacterial clearance [34]. Nevertheless, it has been demonstrated that *S. epidermidis* can evade the death caused by neutrophils through the production of protease SepA, which destroys the antimicrobial peptides (AMPs) produced by the neutrophils [35]. It indicates that the adaptive evolution of *S. epidermidis* towards the neutrophil or the macrophage has generated a beneficial interaction of *S. epidermidis* with neutrophils, where the biofilm is the central strategy to evade the innate immune response [36].

Biofilm-forming isolates can build biofilm in conventional microbiologic culture media, whereas those who do not form biofilm under these conditions are considered non-biofilm-forming isolates [37]. Non-biofilm-forming isolates need to be studied to know if under different conditions they remain with the incapability to form biofilm. Based on the work of Rohde et al. (2005) [20], our group explored the biofilm induction by trypsin in a collection of clinical and commensal non-biofilm forming isolates. We found that a high proportion of commensal *S. epidermidis* isolates from healthy skin stimulated with trypsin induced in-vitro biofilm formation, and this biofilm turned the bacteria into being resistant to ciprofloxacin [37]. This finding is relevant because it supports Rohde et al.’s reports and confirms that commensal *S. epidermidis* isolates are potentially biofilm formers due to the trypsin induction. However, the problem on that scope is that trypsin is not a conventional protein present in inflammatory environments during the establishment of an infection. Now in this work, we extend this hypothesis towards neutrophil proteases based on their relevance to understanding the infection mechanism of non-biofilm-forming isolates.

## 2. Results

### 2.1. Genotypic Characterization and Induction of Biofilm in Non-Biofilm-Forming Staphylococcus Epidermidis Isolates

In order to select the *S. epidermidis* isolates to work with, we first analyzed some genotypic and phenotypic characteristics of *S. epidermidis* isolates. In the genotypic analysis we searched for the *sepA, esp*, and *ecpA* genes that code for external proteases, and for genes that participate in the formation of biofilm including *aap* as a protein-dependent biofilm, and *icaA*, and *icaD* as a PNAG/PIA-dependent biofilm.

A total of eighty-one isolates were distributed in eight genotypes (P1–8) and four sub-genotypes (P1A, P2A, P3A, and P4A; Table 1). The P1 genotype included most of the isolates (30 isolates), followed by the P2 (11 isolates) and P4 (10 isolates) genotypes. The P1 genotype includes the *sepA*^+^*, esp*^+^ and *ecpA*^+^ genes that code for external proteases, and also includes the *aap*^+^ gene that participates in the development of a protein-dependent biofilm. Concerning the *ica* genes, only the isolates with the P3 genotype and the P3A sub-genotype (10 isolates) have the *icaA*^+^ and *icaD*^+^ genes. It indicates that most of the isolates cannot produce a PNAG/PIA-dependent biofilm.

Related to the phenotype analysis, all isolates were analyzed with regard to their faculty to induce biofilm under different conditions that have been reported to increase or induce biofilm formation such as glucose, NaCl, and ethanol. The use of 1% glucose was the condition that caused the highest biofilm formation in the isolates (18.5%), the isolates with P3 and P1 genotypes being those that showed the highest induction of biofilm under this treatment. The treatment with 4% ethanol was the condition with the second most induced biofilm generation, where the isolates with P3A and P1 genotypes had the highest (11.1%) induction of biofilm under this condition. Finally, 4% NaCl was the condition with the lowest biofilm formation in the isolates (6.2%). These results show that only a low proportion of the isolates can induce biofilm formation under certain conditions.

After the genotyping and phenotyping characterization of isolates, we selected some isolates to continue the analysis of their biofilm formation under the presence of neutrophils’ protein extract. From all the 81 isolates we selected 53 isolates with *aap*^+^, *icaA*^−^ and *icaD*^−^ genotype and non-biofilm-forming phenotype even under glucose, ethanol, or NaCl induction.

### 2.2. Induction of Biofilm by Neutrophil Protein Extract

Total protein extracts from human neutrophils and keratinocyte cells (human HaCaT cell line) were obtained. A non-biofilm-forming *S. epidermidis* isolate 54HS (healthy skin) was incubated with total protein extracts from neutrophils or keratinocytes. At high concentrations (0.05 to 0.4 mg/mL) of the neutrophil protein extract, no biofilm was detected; however, at lower concentrations (0.005 to 0.02 mg/mL) of neutrophil protein extract, there was a significant induction of biofilm formation (Figure 1A; *p* < 0.05). This event did not occur when the protein extract from keratinocytes was used (Figure 1B). These results show that certain proteins from neutrophils can induce biofilm formation in a non-biofilm-forming *S. epidermidis* isolate.

### 2.3. Induction of Biofilm by Neutrophil Proteases in Commensal and Clinical Isolates

As neutrophil protein extract induced biofilm formation, we tested the effect of different concentrations of specific neutrophil proteases including cathepsin G, proteinase-3, cathepsin B, and metalloproteinase-9 (MMP-9), tested in commensal and clinical isolates (53 total isolates). Proteinase-3 was the protease that induced biofilm formation in most of the isolates (86.4% of commensal isolates and 38.7% of clinical isolates). The other neutrophil proteases tested induced biofilm formation in percentages between 19.6% and 40.9% of the isolates (Table 2).

When we grouped the commensal and clinical isolates based on their isolation source, we observed that proteinase-3 was the neutrophils’ enzyme that induced biofilm formation (proteinase-3-induced biofilm) in most of the isolates from healthy conjunctiva (HC), healthy skin (HS), ocular infection (OI), and prosthetic joint infection (PJI) (*p* < 0.05 for isolates of HC; Table 3). About the other proteases, cathepsin B was the enzyme with the lowest number of isolates to induce biofilm.

### 2.4. Role of Neutrophil’s Protease in Biofilm Formation

Biofilm formation kinetics was performed to know the participation of neutrophil’s proteases. When we used different concentrations of proteinase-3 enzyme, we found that biofilm formation is dependent on the concentration of proteinase-3 in the isolate 54HS (Figure 2). This effect was also observed in isolates 2HC in the presence of cathepsin G (Appendix A).

Four isolates from different sources were exposed to proteinase-3 to form a proteinase-3-induced biofilm. The biofilm formation kinetics of the proteinase-3-induced biofilm was similar in all four isolates, having an initial adhesion phase of 6 h. Later, the cell aggregation step showed the maximum absorbance up to 18 h in the commensal isolate 2HC and up to 12 h in the clinical isolates (OI and PJI). After 12 h of incubation, the proteinase-3-induced biofilm remained constant in the clinical isolates, but there was a decrease of biofilm in the 2HC and 54HS isolates. In all cases, biofilm formation only occurred in the presence of proteinase-3 (Figure 3).

From the biofilm formation kinetics results, we determined the incubation times required for the final initial adhesion stage of biofilm formation (start of cell aggregation stage) finding that it happens at 6 h of culture and that the full cell aggregation occurs at 12 h of culture. When proteinase-3 was added after 6 h of culture, the biofilm formation kinetics showed a significant decrease in the absorbance in comparison to the biofilm formation kinetics when the proteinase-3 was added since the beginning of the culture in the isolates 54HS and 50OI (Figure 4A,B) and 2HC (Appendix A). When proteinase-3 was added after 10 h (before full cell aggregation), there was no increase in absorbance in the clinical 50OI nor in the commensal 54HS isolates (Figure 4A,B) nor 2HC (Appendix A). These results suggest that proteinase-3 is required for the beginning but also at the end of the cell aggregation stage of biofilm formation.

To support this asseveration, we used the phenylmethylsulfonyl fluoride (PMSF) inhibitor of proteinase-3. Adding PMSF inhibitor at 6 h of culture, there was no increase in the absorbance if compared with the control proteinase-3-induced biofilm without PMSF inhibitor (*p*< 0.05), meanwhile the addition of PMSF after 10 h of culture stops the development of biofilm (Figure 5). This result supports the previous affirmation, that the exogenous proteinase-3 participates in the cell aggregation stage.

### 2.5. Protease-Induced Biofilm in a Mouse Model

A catheter inoculated with the *S. epidermidis* isolate 54HS and proteinase-3 was inserted into the back of a mouse for seven days. After the incubation period, the colony-forming unit (CFU/mL) obtained from the catheter inoculated with *S. epidermidis* isolate 54HS and proteinase-3 was higher than that from the catheter inoculated only with *S. epidermidis* isolate 54HS without proteinase-3 (*p* < 0.05).

The CFU/mL recovered from the catheter inoculated with *S. epidermidis* isolate 54HS in the presence of proteinase-3 was statistically higher than those obtained from the catheter inoculated with the control *S. epidermidis* strain RP62A (biofilm-forming strain), and was also higher than those recovered from the catheter inoculated with *S. epidermidis* isolate 54HS and trypsin (Figure 6). These results show that, in-vivo, there is an induction of biofilm formation by proteinase-3.

### 2.6. Biofilm Induction by the Presence of Neutrophils

Challenge tests were performed between *S. epidermidis* isolates (2HC, 54HS and 50OI) and neutrophils to determine the neutrophils’ role in the induction of biofilm formation. In all three isolates, neutrophils induced biofilm formation only when a multiplicity of infection (MOI) of neutrophils—bacteria of 1:0.001 and 1:0.01—were used, in comparison with the control group without neutrophils (*p* < 0.05; Figure 7A). This induction of biofilm formation by neutrophils may be due to neutrophil’s proteases or neutrophil’s eDNA. In order to determine whether proteases or eDNA are involved in biofilm formation, challenge tests were performed between *S. epidermidis* isolates (2HC, 54HS and 50OI) and neutrophils treated with a cocktail of protease inhibitors (Sigma-Aldrich, Merck, State of Mexico, Mexico) or DNAse. Neutrophils-bacteria treated with a cocktail of protease inhibitors had a reduction of biofilm formation with a statistically significant difference compared to untreated neutrophils-bacteria at different MOI of neutrophils:bacteria (Figure 7B and Appendix A); however, in the DNAse treatment a tendency to reduce biofilm formation occurred, but without statistically significant differences. This result suggests that neutrophil’s proteases are involved in biofilm formation, but eDNA may also contribute.

The neutrophil’s DNA can be exposed to the environment by NETs formation and it could be used by some bacteria as support to produce its own biofilm. We investigated whether *S. epidermidis* induces NETs formation or not. So, we co-cultured neutrophils with *S. epidermidis* isolate 54HS or *Staphylococcus aureus* (as positive control of NETs formation). In each experiment, phorbol 12-myristate 13-acetate (PMA) was used as a control for NETs formation. We observed NETs formation under PMA stimulation, and a reduction when DNA was digested with DNAse (Figure 8A). NETs were not induced at low MOI (neutrophils:bacteria, 1:0.01, 1:0.1 and 1:1) in both bacteria (Appendix A). However, with high MOI (1:5 and 1:10) the NETs were induced in both bacteria (Figure 8B and Appendix A). After the analysis of three independent experiments, the percentage of NETs formation is reported and represented in Figure 8C. The neutrophils induced NETs in 1.7% of the cases under basal conditions, but it increased considerably to 97.3% under PMA stimulation, and when neutrophils with PMA stimulation were treated with DNase the NETs formation reduced to 14.3%. *S. aureus* showed 15.8% and 27.1% of NETs when MOI 1:5 and 1:10 were used respectively, meanwhile with *S. epidermidis* it was 8% and 13.8%, respectively. In conclusion, our results demonstrate that *S. epidermidis* can induce NETs formation only when MOI (neutrophils:bacteria) 1:5 and 1:10 are used, suggesting that neutrophil eDNA is not participating in the induction of biofilm formation.

As NETs were not induced with MOI (neutrophils:bacteria) 1:0.001 nor 1:0.01, the neutrophils’ DNA release did not occur, meaning that the eDNA may be of bacterial origin. So then, proteinase-3-induced biofilm from isolate 54HS was treated with DNAse in the absence of neutrophils, and we found that when proteinase-3-induced biofilm is treated with DNAse a significative reduction of biofilm formation occurs if compared with the untreated proteinase-3-induced biofilm (*p*< 0.05). The reduction of biofilm after DNAse treatment was such that it is comparative to that seen in the controls (Figure 9A); this event also occurs with isolates 2HC and 50OI (Appendix A).

Finally, to confirm the participation of the chromosomal DNA released by the bacteria in the biofilm formation, the proteinase-3-induced biofilm of isolate 54HS was treated with DNAse at different times of incubation; at the beginning of incubation (0 h) and at 8 h there was a significant reduction in biofilm formation compared with the untreated condition; meanwhile, this effect did not occur when DNAse was added at 10 or 12 h (Figure 9B) of incubation. The same occurs with isolates 2HC and 50OI (Appendix A). With these results, we show that proteinase-3 induces the release of eDNA from the bacteria at an early time (at the beginning of the initial adhesion stage) and it could be a mechanism of biofilm formation.

## 3. Discussion

Rohde et al. demonstrated that the Aap protein is proteolytically processed to induce biofilm formation in a non-biofilm-forming *S. epidermidis* 1457 strain [20]. Our research group decided to try the effect of trypsin (considered an external protease) over a collection of clinical and commensal isolates, finding that trypsin is a good inducer of biofilm formation in the commensal non-biofilm-forming isolates [37]. In this work, it is stated that neutrophil proteases are also potential inducers of biofilm in isolates that cannot form a biofilm. This result is significant because it emphasizes that commensal *S. epidermidis* could induce biofilm in the presence of neutrophils promoting the survival of *S. epidermidis*.

The genotyping analysis showed that most of the non-biofilm-forming isolates have the *aap*^+^ gene, which encodes to Aap protein, and are *icaA*^−^ and *icaD*^−^, indicating that these isolates are unable to produce an PNAG/PIA-dependent biofilm. The frequency in the presence of the *aap*^+^ gene in the isolates suggests that the protease-induced biofilm is a protein-dependent biofilm. The genotype of the *S. epidermidis*’ extracellular proteases also shows that the isolates have a *sepA*^+^ genotype. It has been reported that protease SepA is involved in the processing of Aap for biofilm formation [21]; however, the isolates did not form biofilm in conventional media under either of the different conditions we tested, suggesting that SepA protease is not participating. Moreover, a proportion of isolates showed biofilm induction under different conditions, such as the presence of glucose (18.5%), ethanol (6.2%), or NaCl (11.1%). However, these isolates were excluded from the study to avoid background in the induction of biofilm by the presence of neutrophil proteases.

The total protein extract from HaCaT keratinocyte cells did not induce biofilm in non-biofilm-forming *S. epidermidis* isolates. It suggests that keratinocyte proteases cannot induce biofilm in this bacterium, probably because *S. epidermidis* is an inhabitant of the skin and has continuous contact with keratinocytes, which evolutionarily has generated a mutually beneficial relationship between bacteria and the host’s cells [38] avoiding infection by biofilm. Anti-leukoprotease (ALP), also known as mucous protease inhibitor or secretory leukoprotease inhibitor, is constitutively expressed in keratinocytes; ALP exhibits antimicrobial activity against several human skin associated microorganisms such as *P. aeruginosa, S. aureus* and *S. epidermidis*. In that way, ALP represents a major soluble serine protease inhibitor and an antimicrobial agent that contribute to the high resistance of epidermis against proteolysis and infections [39]. In contrast, neutrophil protein extract could induce biofilm, suggesting that these cell proteases are involved in this process. We tested four types of neutrophil proteases, and proteinase-3 induced biofilm in a larger number of clinical and commensal isolates compared to the other three proteases. Interestingly, the commensal isolates had a greater capability to form a proteinase-3-induced biofilm than the clinical isolates. It suggests that the commensal *S. epidermidis* that inhabit the skin have the potential to be infective by biofilm formation after having entered inside the body through a medical device; in this situation neutrophil proteases can induce biofilm and thus protect *S. epidermidis* from the host’s immune system.

Clinical isolates from PJI were the most refractive to the biofilm induction by neutrophil proteases; this also occurs with the biofilm induction by trypsin [37]. We do not know what could be happening in these isolates, but it should be considered that these PJI isolates have a different mechanism of infection. It is known that in a biofilm, there is a heterogeneous bacterial community composed of active bacteria as well as bacteria with low metabolism known as dormant bacteria [40]; these bacteria in this state are more tolerant to the antibiotics and may be associated with their persistence [41]. The *S. epidermidis* isolates from PJI have an infective mechanism due to a reduced metabolic activity and slow growth (dormant bacteria), which helps to evade the immune response [16,27,28], and to wait for a suitable condition to trigger its infective process [42].

We found that proteinase-3 is involved in the initial adhesion for biofilm formation, as well as in the cell aggregation stage. In the initial adhesion to abiotic support, AtlE, Aee proteins and teichoic acids are involved. AtlE needs to be proteolytically processed to be activated [23], thus it is likely that neutrophil proteases break AtlE to activate its function. AtlE is a bifunctional protein as it participates in the cell adhesion and in the cell lysis; the cell lysis helps to release the bacterial eDNA that serves as a support for the formation of biofilm [18,22]. We show that the bacterium releases eDNA by proteinase-3 activity in an early time (from 0 at 8 h) for biofilm formation, suggesting that AtlE could be involved in the lysis of the bacteria, as already suggested by Qin Z et al. (2007) [43].

Rohde et al. (2015) proposed that Aap protein is involved in biofilm formation, and the mechanism is the proteolytic processing (endogenous or exogenous) of the Aap protein. This proteolytic processing can be carried out by various proteases such as trypsin, cathepsin G, elastase [20], and SepA from *S. epidermidis* [21]. The finding that different proteases are capable of breaking down the Aap protein suggests that the *S. epidermidis*’ Aap protein has various proteolytic cleavage sites for several proteases. This fact leads to the thinking that, during the evolution of *S. epidermidis*, the Aap protein modified its proteolytic sites to adapt to different proteolytic systems, suggesting an evolutionary survival mechanism of *S. epidermidis* to settle in various human locations [38]. On the other hand, we do not discard the possibility that other surface proteins not yet described could be proteolytically processed and could participate in the biofilm formation of *S. epidermidis*, such as the AtlE protein.

The results in the biofilm formation kinetics demonstrated that proteinase-3 participates in the cell aggregation stage, since the addition of proteinase-3 inhibitor at the time of cell aggregation significatively reduced biofilm formation, suggesting that proteinase-3 (proteolytically) activates Aap for cell aggregation, as shown by Rohde et al. (2015) [20]. We do not rule out that cathepsin G, cathepsin B and MM-9 may have the same effect, and further experiments are underway.

Regarding the experiments carried out with neutrophils for the induction of NETs, in the case of *P. aeruginosa* the induction of NETs is beneficial for the bacteria because the DNA released in the NETs is used as DNA-dependent support for the increase of the *P. aeruginosa*’s biofilm [25,26]. It has also been described that *P. aeruginosa* increases its biofilm formation on contact lenses, as *S. epidermidis*, *S. aureus, Serratia marcescens*, and *Stenothophomonas maltophilia* do [44], suggesting a common mechanism in these bacteria. In contrast, according to our results, in the case of non-biofilm forming *S. epidermidis*, biofilm formation is due to the release of the bacteria’s own DNA and not by NET’s.

Here, we demonstrate that the addition of neutrophil proteases is enough to induce biofilm in non-biofilm-forming isolates, that proteinase-3 is important for cell aggregation as well as for the initial adhesion stage and, finally, that neutrophils’ proteases (we suggest proteinase-3, cathepsin B, and MM-9) induce biofilm formation in non-biofilm-forming *S. epidermidis*. Our results suggest that neutrophils do not generate NETs in the presence of non-biofilm-forming *S. epidermidis*, but instead produce proteases, such as proteinase-3, which activates AtlE (proteolytically) to promote the release of bacterial eDNA that will be used in the adhesion of *S. epidermidis* to an abiotic surface. Besides, proteinase-3 could also act proteolytically over Aap for cell aggregation and thus induce the formation of biofilm.

*S. epidermidis* uses a passive defense to evade destruction by the neutrophils, because *S. epidermidis* is a less aggressive pathogen than *S. aureus* [35]. Besides, *S. aureus* secretes the extracellular adherence protein (Eap) and homologs EapH1 and EapH2 that potently inhibit the activity of neutrophils elastase, proteinase-3 and cathepsin G [45]. It confirms that *S. epidermidis* goes unnoticed in the early stages of infection, leading to slow growth and dormancy for monitoring the environmental conditions for subsequent adaptation [46]. It is necessary to remember that the *S. epidermidis* infection on medical devices occurs in late periods (months) [47,48]. It may explain, in part, the in-vivo results observed, where in non-biofilm-forming *S. epidermidis* isolates that were not treated with proteinase-3, after 7 days, bacterial biofilm was recovered in a lower quantity from the catheter than those treated with proteinase-3, suggesting a slow growth until waiting for the optimal conditions (endogenous mouse protease) to form the biofilm.

The understanding of in-vivo infection from a contaminated medical device is very complex, although the immune system, specifically neutrophils, helps to avoid infection with success. It also depends on the type of microorganism present; perhaps a virulent bacterium quickly stimulates an immune response leading to a battle between the immune cells and the microorganism, giving, as a result, the elimination of the bacteria. In the case of a commensal microorganism inhabiting some part of the human body, it may have the capacity to know the host’s innate immune mechanisms and how to take control and take advantage of it for its survival. Thus, *S. epidermidis*, with few weapons to infect, has evolved, benefiting from the immune system by generating its biofilm under adverse conditions—one of them is the presence of neutrophils.

## 4. Materials and Methods

### 4.1. Isolates

*S. epidermidis* isolates were obtained from patients with ocular infection (OI; *n* = 8), healthy skin (HS; *n* = 12), healthy conjunctiva (HC; *n* = 10), and with prosthetic joint infections (PJI; *n* = 23). All isolates were reported previously by Martínez-García et al. (2019) [37] and were classified as in-vitro non-biofilm-forming strains, determined by Christensen’s standard method [49].

### 4.2. Genomic DNA Extraction

Bacterial cells were grown overnight in tryptic soy broth (TSB; Sigma-Aldrich, Merck, Toluca, Mexico), harvested by centrifugation, and resuspended in 200 μL of lysis solution (20% sucrose, 10 mM Tris-HCl, pH 8 and 10 µg/mL lysozyme). Cells were incubated at 37 °C for 40 min and added 200 μL of Whinston solution (2% Triton X-100, 1% SDS, 10 nM NaCl, 10 mM Tris-base, pH 8.0, and 1 mM EDTA). DNA was extracted with phenol/chloroform/isoamyl alcohol (25:24:1). DNA was subsequently precipitated with one volume of isopropanol and desalted by the addition of two volumes of 70% ethanol. Finally, DNA was resuspended in sterile distilled water free of DNases.

### 4.3. Amplification of Genes by PCR

The *app*, *sepA*, *esp*, *ecpA*, *icaA*, and *icaD* genes were amplified using the primers listed in Table 4. PCR amplifications were performed using 1 μL of DNA template (100 ng), 1× buffer, 1 mM MgCl_2_, 5 mM of each dNTPs, 1 U of Taq DNA polymerase (Invitrogen, Thermo Fisher Scientific, Waltham, MA, USA), and 0.2 μM of each specific primer. PCR conditions were as follows: 30 cycles of 30 s at 92 °C, 40 s at 60 °C, and 30 s at 72 °C. PCR products were analyzed on agarose gels.

### 4.4. Biofilm Formation

According to Christensen’s standard method, non-forming biofilm isolates were treated with different substances to determine the capacity to induce biofilm [49]. Overall, the procedure was as follows: isolates were inoculated in TSB (Sigma-Aldrich) and incubated for 24 h at 37 °C. They were then inoculated into 96-well tissue culture plates (Nunc, Thermo Fisher Scientific) in TSB medium (1:200 dilution). To the different treatments, TSB medium was mixed with the following substances at different dilutions: 1% glucose, or 4% NaCl or 4% ethanol, or protein extract from neutrophil cells, or protein extract of HaCat cells at 0.4, 0.2, 0.1, 0.05, 0.02, 0.01, 0.005 mg/mL. For human neutrophil proteases: 0.1, 1, 10, or 100 ng/mL of cathepsin G; 0.25, 2.5 or 25 ng/mL of proteinase-3; 0.1, 1, 10, or 100 ng/mL of cathepsin B and 0.01, 0.1, 1, or 10 ng/mL of metalloproteinase-9 (MMP-9); as control 200, 2000, or 20,000 ng/mL of trypsin (Gibco, Thermo Fisher Scientific). Plates were incubated for 24 h at 37 °C. Following the incubation period, the plates were washed vigorously with 1× phosphate-buffered saline (PBS), dried for 30 min at 55 °C, and stained with 0.5% (*w*/*v*) crystal violet solution. After staining, the plates were washed with 1× PBS. The absorbance (A_492_) of adhered, stained cells was measured using a Multiskan GO Microplate spectrophotometer (Thermo Fisher Scientific). The average A_492_ values were calculated for all tested isolates. *S. epidermidis* RP62A and *S. epidermidis* ATCC 12228 were used as positive and negative control strains, respectively, and all tests were performed in triplicate and repeated three times. A cut-off value (A_492c_) was established. It is defined as three standard deviations (SD) above the mean A_492_ of the negative control: A_492c_ = average A_492_ of negative control + (3 × SD of negative control). The final A_492_ value of a tested strain was expressed as the average A_492_ value of the strain reduced by A_492c_ value (A_492_ = average A_492_ of a strain—A_492c_). The A_492c_ value was calculated for each microtiter plate separately.

Since activated neutrophils release many proteases, their specific effect is not easy to study. Based on the work of Rohde et al. (2005) [20], where they used cathepsin G, elastase, trypsin and, based on the extracellular proteolytic system of *S. epidermidis* that expresses a cysteine protease (EcpA), a serine protease (Esp) and a metalloprotease (SepA), we used neutrophil proteases with these characteristics. Therefore, we selected for this work two representative serine proteases (cathepsin G and proteinase-3), a cysteine protease (cathepsin B), a metalloprotease (metalloprotease-se-9, MMP-9), and trypsin as a control.

### 4.5. Biofilm Formation Kinetics

The biofilm determination procedure was performed as described above. For the biofilm formation kinetics, 96-well tissue culture plates (Nunc, Thermo Fisher Scientific) with removable wells were used, and at each time (every 8 h) of the kinetics, a wells column was removed from the plate for the analysis.

After determining the biofilm formation kinetics of the isolates, the biofilm formation kinetics were performed in the presence of proteinase-3 at different concentrations (0.25, 2.5 or 25 ng/mL; Gibco, Thermo Fisher Scientific) added to the medium, as well as the biofilm formation kinetics in the presence of proteinase-3 plus 5 mM phenylmethylsulfonyl fluoride (PMSF; Sigma-Aldrich) used as proteinase-3 inhibitor. As controls, trypsin and trypsin inhibitor from bovine pancreas (Sigma-Aldrich) were used.

### 4.6. Mouse Model of Catheter Infection

Female Balb/c mice were used in a model of subcutaneous implanted device-related infection according to Sander et al. (2012) [50]. This study was carried out following the recommendations of the bioethics review board of “Escuela Nacional de Ciencias Biológicas-IPN.” The mice were weighed and anesthetized by intraperitoneal injection with filocain (100 mg/g of weight). The hair was removed from the back using electric hair clippers, and a small incision was made. Then 1 cm of a sterile 14-gauge teflon intravenous catheter (Exel International) was inserted, and the incision was sutured. Approximately 1.5 × 10^8^ CFU in 20 μL of sterile PBS of either an individual bacterial strain or the strain with 25 ng/mL of proteinase-3, or 2 µg/mL of trypsin were injected into the inserted catheter. After 7 days post-infection, the animals were sacrificed. The catheters were removed, and each one was put in 1 mL of PBS 1 × and sonicated at 200 Hz for 5 min twice, and CFU/mL were determined. Three individual experiments were done using three mice each time. All animal experiments were carried out following the National Institutes of Health guidelines for the care and use of laboratory animals, and the protocol was approved by the ethics review board of our institution.

### 4.7. Biofilm Induced by Neutrophils

Neutrophils were isolated from peripheral whole blood from three healthy donors after centrifugation at 500× *g* during 30 min on Polymorphprep density gradient (Alere Technologies AS, Oslo, Norway). Then, the neutrophils were washed with supplemented culture medium (RPMI 1640, 2 mM L-glutamine) and finally resuspended in supplemented RPMI containing 10% of fetal bovine serum (FBS). The cells were counted, the viability determined by trypan blue exclusion being higher than 95%. Finally, the cells were adjusted at 1 × 10^6^ per milliliter.

Neutrophils (1 × 10^6^ cells/mL) were placed into 96-well plates (Nunc, Thermo Fisher Scientific) with *S. epidermidis* at multiplicity of infection (MOI; neutrophils:bacteria) 1:0.001, 1:0.01, 1:0.1 or 1:1 in RPMI supplemented with 10% FBS. The culture of neutrophils without *S. epidermidis* was used as control. Then, the biofilm determination was performed as previously described.

### 4.8. Neutrophils Extracellular Traps (NETs)

For NETs analysis 0.25 × 10^6^ neutrophils/well were added to Lab-Tek chamber slides (Nunc, Thermo Fisher Scientific) and *Staphylococcus aureus* or *S. epidermidis* were added at the multiplicity of infection (MOI; neutrophils:bacteria) 1:0.01, 1:0.1, 1:1, 1:5, and 1:10. In this experiment PMA (50 ng/mL, Sigma-Aldrich, Merck) stimulation was included as positive control and medium as negative control. In the last 1 h incubation 100 Units of DNase I (Qiagen GmbH, Hilden, North, Germany) was added to the PMA stimulation to determine that NETs are made of DNA.

After 4 h incubation at 37 °C and 5% CO_2_, the cultures were fixed for 10 min in 4% paraformaldehyde, followed by DNA staining with Sytox Green (Life Technologies, Eugene, OR, USA, dilution 1:100 in PBS) for 10 min. The supernatants were discarded, and the cell preparation was mounted with Prolong Diamond Antifade Mountain (Molecular probes, Life Technologies) and was left overnight at room temperature. The slides were analyzed with a Carl Zeiss fluorescence microscope at 40× and 100×, and the results reported as the percentage of NETs forming cells.

### 4.9. Statistical Analysis

Fisher’s exact test was conducted for proportion analysis, and to analyze biofilm formation, two-way ANOVA and Tukey’s test were conducted. These analyses were carried out with the software GraphPad Prism version 7.0.

## 5. Conclusions

Our results suggest that commensal isolates with *aap*^+^, *sepA*^+^, *icaA*^−^ and *icaD*^−^ genotypes, which have been previously established as non-biofilm formers under described conditions (glucose, NaCl, and ethanol), can produce biofilm in the presence of neutrophil proteases such as cathepsin G, cathepsin B, proteinase-3, MM-9 and in the presence of neutrophil cells. The biofilm induction mechanism depends on the neutrophils, which do not generate NETs but produce proteases such as proteinase-3, possibly affecting the bacterial AtlE protein promoting the release of bacterial DNA as a support for the initial adhesion step of biofilm. Proteinase-3 subsequently participates in the cell aggregation step, leading to biofilm formation.

## Figures and Tables

**Figure 1 ijms-23-04992-f001:**
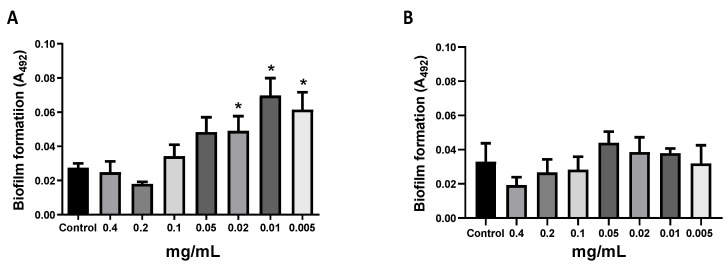
Biofilm formation induced by neutrophil protein extract. Total proteins were obtained from neutrophils and HaCaT cells, and the protein concentration was determined by Bradford. A bacterial culture from overnight incubation was diluted 1:200 and cultured with different concentrations of the neutrophil’s protein extract (**A**) and HaCaT protein extract (**B**) and the amount of biofilm formation was determined by the Christensen’s et al. method. Asterisk * indicates a significant difference (*p* < 0.05) concerning the control (bacteria without protein extract). The statistical analysis was performed using a one-way ANOVA with a Tukey’s test.

**Figure 2 ijms-23-04992-f002:**
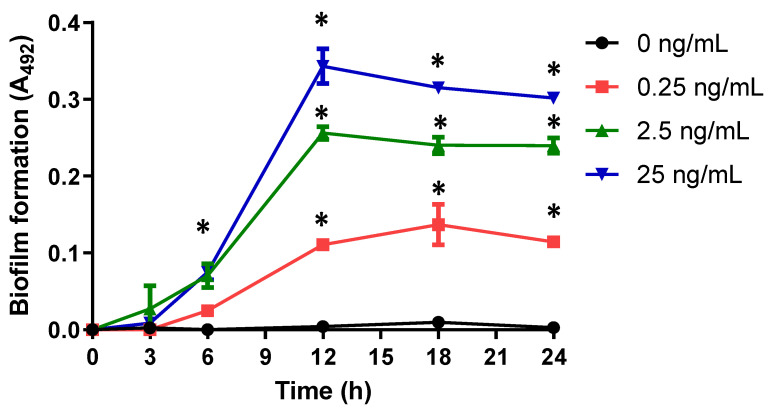
Kinetics of biofilm formation with different concentrations of proteinase-3 in isolate 54HS. Proteinase-3 was added at different concentrations from the beginning of the biofilm formation kinetics. At different times, the biofilm formation was determined by the method of Christensen et al. in isolate 54HS (healthy skin). Asterisk * indicates a significant difference (*p*< 0.05) concerning the control (bacteria without proteinase-3). The statistical analysis was performed using a one-way ANOVA with Tukey’s test.

**Figure 3 ijms-23-04992-f003:**
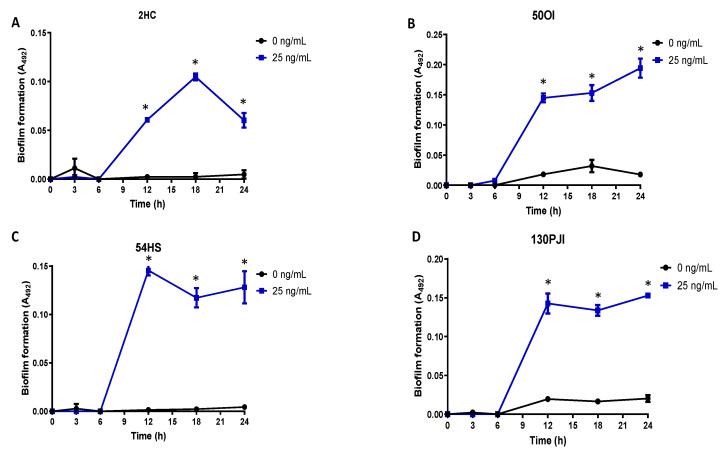
Kinetics of biofilm formation in non-biofilm-forming isolates. Four non-biofilm-forming isolates from different isolation sources ((**A**) HC, healthy conjunctiva; (**B**) OI, ocular infection; (**C**) HS, healthy skin and (**D**) PJI, prosthetic joint infection) were induced for biofilm by proteinase-3 treatment. Asterisk * indicates significant difference (*p* < 0.05) concerning the control (bacteria without protease). The statistical analysis was performed using a one-way ANOVA with Tukey’s test.

**Figure 4 ijms-23-04992-f004:**
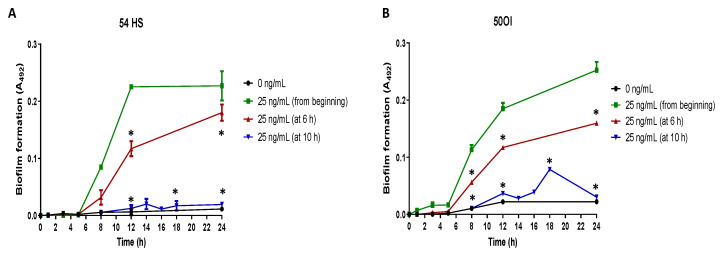
Proteinase-3 addition at different stages of biofilm formation. Biofilm forming kinetics of a commensal (HS, healthy skin, panel (**A**)) and a clinical (OI, ocular infection, panel (**B**)) non-biofilm-forming isolate was done in the presence of proteinase-3 added at 6 or 10 h. The abundance of biofilm was determined according to Christensen et al. sampling at different points of the kinetics. Asterisk * indicates a significant difference (*p* < 0.05) concerning the bacteria with proteinase-3 since the beginning of the culture. The statistical analysis was performed using a one-way ANOVA with a Tukey’s test.

**Figure 5 ijms-23-04992-f005:**
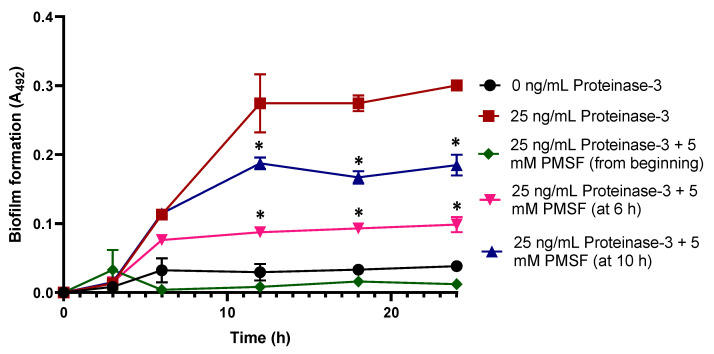
Assay of protease inhibition in the biofilm formation kinetics in isolate 54HS. Proteinase-3 was added to the kinetics from the beginning, and in addition 5 mM PMSF (a proteinase-3 inhibitor) was incorporated to the kinetics at 6 or 10 h after the starting point of culture in the isolate 54HS. The abundance of biofilm was determined according to Christensen et al. sampling at different points of the kinetics. Asterisk * indicates a significant difference (*p* < 0.05) compared to the bacteria with proteinase-3 since the beginning. The statistical analysis was performed using a one-way ANOVA with Tukey’s test.

**Figure 6 ijms-23-04992-f006:**
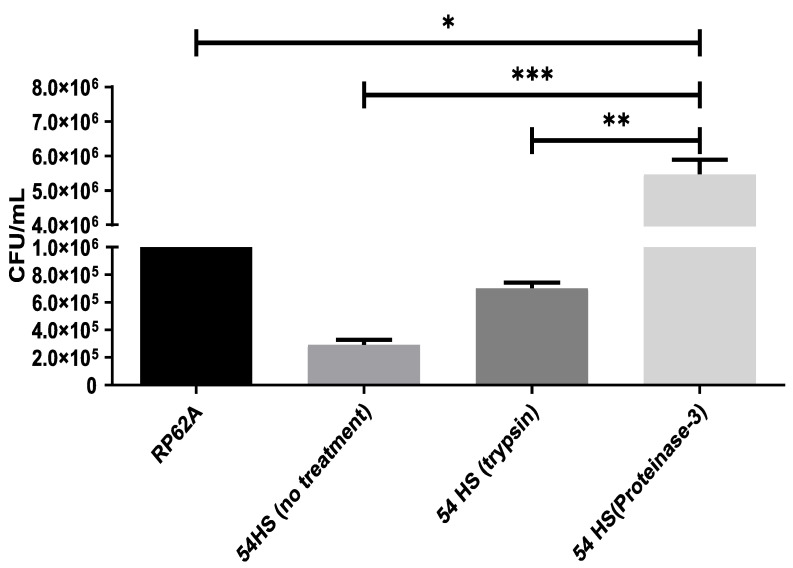
In-vivo biofilm formation by proteinase-3. Catheters were inoculated with the healthy skin non-biofilm-forming isolate 54HS or with the RP62A strain (control, biofilm-forming phenotype). In one catheter 2 µg/mL trypsin was added, and in the other 25 ng/mL proteinase-3 was added. Subsequently, the catheters were inoculated in the back of mice for 7 days. The quantity of viable bacteria attached to the catheters was determined. Asterisks indicate a significant difference (* *p* < 0.05, ** *p* < 0.01, and **** p* < 0.001). The statistical analysis was performed using a one-way ANOVA with Tukey’s test.

**Figure 7 ijms-23-04992-f007:**
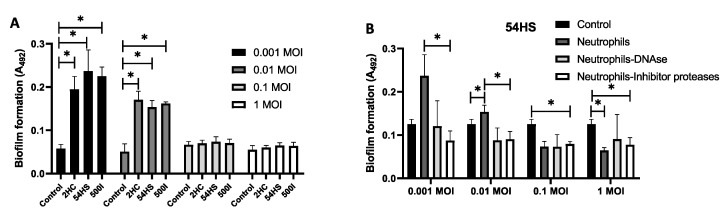
Neutrophils induce biofilm in non-biofilm-forming isolates. (**A**) Neutrophils (600 cells/mL) in the presence of different MOI (neutrophils:bacteria) of non-biofilm-forming isolates were grown in wells with RPMI 10% FBS for 24 h. Biofilm formation was determinate by Christensen’s et al. method. (**B**) Neutrophils (600 cells/mL) were cultured in the presence of different MOI (neutrophils:bacteria) of non-biofilm-forming 54HS isolate (healthy skin) under different conditions: bacteria (control); neutrophils with bacteria (neutrophil); neutrophils with bacteria treated with 2 U DNAse I (Neutrophils-DNAse); neutrophils with bacteria treated with 1X cocktail of proteases inhibitors (Neutrophil-inhibitor proteases). After 24 h of culture the biofilm formation was determinate by the method of Christensen et al. Asterisk * indicates a significant difference (*p* < 0.05). The statistical analysis was performed using a one-way ANOVA with Tukey’s test.

**Figure 8 ijms-23-04992-f008:**
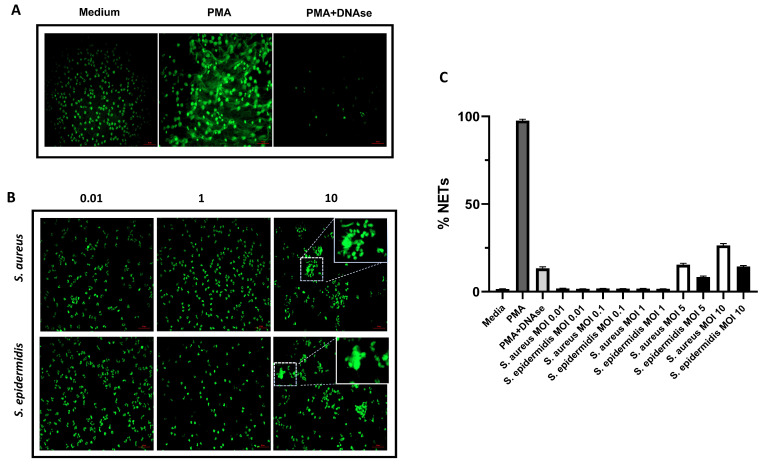
NETs formation. (**A**) Control conditions for NETs formation: neutrophils grown in media without bacteria (media); neutrophils treated with phorbol 12-myristate 13-acetate (PMA) for induction of NETs; neutrophils treated with PMA and DNAse (PMA + DNAse 100 U). (**B**) Induction of NETs in neutrophils (0.25 × 10^6^ cells /mL) grown in the presence of different MOI (neutrophils:bacteria, 1:0.01, 1:1, 1:10) of non-biofilm-forming *S. epidermidis* isolate 54HS or *S. aureus* USA300 strain, after 4 h of culture. (**C**) Percentage of NETs formation of panel A and B assay, considering the number of cells counted as 100%. Scale bar in red: 50 µm.

**Figure 9 ijms-23-04992-f009:**
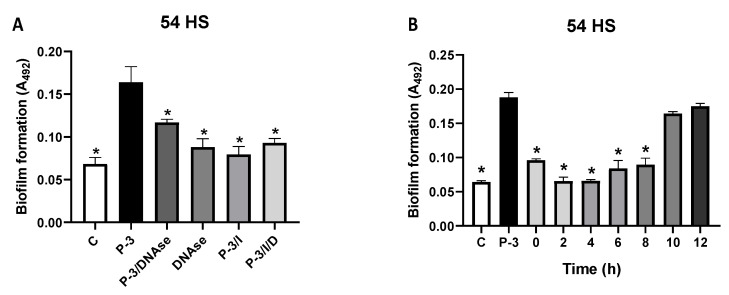
Effect of DNAse in proteinase-3-induced biofilm. (**A**) Bacteria without treatment with proteinase-3, C; bacteria treated with proteinase-3 at 25 ng/mL (P-3); bacteria treated with proteinase-3 and with 2 U DNAse I (P-3/DNAse); bacteria treated with 2 U DNAse I (DNAse); bacteria treated with proteinase-3 and with 5 mM PMSF inhibitor (P-3/I); bacteria treated with proteinase-3, and 5 mM PMSF inhibitor and 2 U DNAse I (P-3/I//D). (**B**) Bacteria without treatment with proteinase-3, C; bacteria treated with proteinase-3 at 25 ng/mL (P-3); bacteria treated with proteinase-3 more 2 U DNAse I at different times (0, 2, 4, 6, 8, 10, and 12 h). The abundance of biofilm was determined according to Christensen et al. Asterisk * indicates a significant difference (*p* < 0.05) to the P-3. The statistical analysis was performed using a one-way ANOVA with Tukey’s test.

**Table 1 ijms-23-04992-t001:** Genotypic and phenotype characteristics in *S. epidermidis* with non-biofilm forming phenotype.

Genotype	Number Isolates	*aap*	*sepA*	*Esp*	*ecpA*	*icaA*	*icaD*	Glucose 1%	NaCl 4%	Ethanol 4%
P1	30	+	+	+	+	−	−	8 (26.6)	2 (6.6)	5 (16.6)
P1A	1	−	+	+	+	−	−	0	0	0
P2	11	+	−	−	−	−	−	1 (9.1)	3 (27.3)	1 (9.1)
P2A	4	−	−	−	−	−	−	1 (25)	0	0
P3	9	+	+	+	+	+	+	5 (55.5)	0	1 (11.1)
P3A	1	−	+	+	+	+	+	0	0	1 (100)
P4	10	+	+	−	−	−	−	0	0	1 (10)
P4A	4	−	+	−	−	−	−	0	0	0
P5	5	+	+	−	+	−	−	0	0	0
P6	2	+	+	+	−	−	−	0	0	0
P7	2	+	+	+	−	−	+	0	0	0
P8	2	−	+	−	+	−	−	0	0	0
Total	81	69	65	45	48	10	12	15 (18.5)	5 (6.2)	9 (11.1)

**Table 2 ijms-23-04992-t002:** Biofilm formation in *S. epidermidis* isolates in the presence of neutrophils’ proteases.

Isolates (*n* = 53)	Cathepsin G *n* (%)	Proteinase-3 *n* (%)	Cathepsin B *n* (%)	MMP-9 *n* (%)
Commensal isolates *n* = 22	8 (36.4)	19 (86.4) *	7 (31.8)	9 (40.9)
Clinical isolates *n* = 31	6 (19.6)	12 (38.7)	7 (22.6)	8 (25.8)

* Statistical difference when compared with other proteases according to Fisher analysis. Cathepsin G at 0.1, 1, 10, or 100 ng/mL; proteinase-3 at 0.25, 2.5 or 25 ng/mL; cathepsin B at 0.1, 1, 10, or 100 ng/mL; and metalloproteinase-9 (MMP-9) at 0.01, 0.1, 1, or 10 ng/mL. The isolates showed biofilm induction at different concentrations of the tested proteases. For example, some isolates produced biofilm only at the lowest concentrations of proteases, others induced biofilm up to intermediate concentrations, and others did so at all tested concentrations. The table shows the total number of isolates that induced biofilm by proteases.

**Table 3 ijms-23-04992-t003:** Protease-induced biofilm in *S. epidermidis* isolates from different isolation source.

Source of Isolation (*n* = 53)	Cathepsin G *n* (%)	Proteinase-3 *n* (%)	Cathepsin B *n* (%)	MMP-9 *n* (%)
healthy conjunctiva (HC), *n* = 10 (18.8%)	4 (40)	10 (100) *	3 (30)	5 (50)
healthy skin (HS), *n* = 12 (22.6%)	4 (33.3)	9 (75)	4 (33.3)	4 (33.3)
ocular infection (OI), *n* = 8 (15%)	5 (52.5%)	7 (87.5)	3 (37.5)	5 (52.5)
prosthetic joint infection (PJI), *n* = 23 (43.4%)	1 (4.3)	5 (21.7)	4 (17.4)	3 (13)

* Statistical difference when compared with other proteases according with Fisher analysis. Cathepsin G at 0.1, 1, 10, or 100 µg/mL; proteinase-3 at 0.25, 2.5 or 25 ng/mL; cathepsin B at 0.1, 1, 10, or 100 ng/mL; and metalloproteinase-9 (MMP-9) at 0.01, 0.1, 1, or 10 ng/mL. The isolates showed biofilm induction at different concentrations of the tested protease. For example, some isolates produced biofilm only at the lowest concentrations of proteases, others induced biofilm up to intermediate concentrations, and others did so at all tested concentrations. The table shows the total number of isolates that induced biofilm by proteases.

**Table 4 ijms-23-04992-t004:** Primer sequence.

Gene	Sequence (5′→3′)
*sepA*	Fw:CCAGGGAGCAGCCTCGATGAAGAATTTTTCTAAATTC Rv:GCAAAGCACCGGCCTCGTTACTACACGCCAACAC
*esp*	Fw: CCAGGGAGCAGCCTCGATGAAAAAGAGATTTTTATC Rv:GCAAAGCACCGGCCTCGTTACTGAATATTTATATCAGG
*ecpA*	Fw: CCAGGGAGCAGCCTCGATGAAGAAAAAATTAAG Rv: GCAAAGCACCGGCCTCGTTAATAACCATAAATTGATG
*icaA*	Fw: TCTCTTGCAGGAGCAATCAA Rv: AGGCACTAACATCCAGCA
*icaD*	Fw: ATGGTCAAGCCCAGACAGAG Rv: CGTGTTTTCAACATTTAATGCAA
*aap*	Fw: AGAAACAAGCTGGTCAAG Rv: CTGCGTAGTTAAGAAAATC

## Data Availability

Not applicable.

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
