# Peer review of "Low Concentration of the Neutrophil Proteases Cathepsin G, Cathepsin B, Proteinase-3 and Metalloproteinase-9 Induce Biofilm Formation in Non-Biofilm-Forming *Staphylococcus epidermidis* Isolates"

_ijms, 2022, doi:10.3390/ijms23094992_

Round 1

Reviewer 1 Report

The authors tested Cathepsin 27 G, cathepsin B, proteinase-3, and metalloproteinase-9 (MMP-9) from neutrophils on the biofilm induction in commensal (skin isolated) and clinical non-biofilm-forming Staphylococcus epidermidis isolates.

The results suggest that low level of neutrophils, in the presence of commensal non-biofilm forming S. epidermidis, significantly induced the biofilm formation, but do not generate neutrophil extracellular traps formation. The effect of neutrophils inducing biofilm is the production of proteases, and proteinase-3 releases bacterial DNA at the initial adhesion, favouring cell aggregation and subsequently leading to biofilm formation.

The study adds the novel knowledges and is worth for publication. However, the points below need to be addressed:

  1. Title is a bit misleading, inducing biofilm formation is only happened in low concentrations. Suggest adding “Low concentration of” in front of neutrophil…. Staphylococcus epidermidis need to be italic “Staphylococcus epidermidis”.

Suggested new title: “Low concentration of neutrophil proteases Cathepsin G, cathepsin B, proteinase-2 and metalloproteinase-9 induce biofilm formation in non-biofilm-forming Staphylococcus epidermidis isolates”.

Line 174 – 176. “At high concentrations (0.4 to 0.05 mg/mL) of the neutrophil protein extract, no biofilm was detected, however, at lower concentrations (0.02 to 0.005 mg/mL) of neutrophil protein extract there was a significant induction of biofilm formation.”

  1. Abstract: Line 44. Please spell MOI: multiplicity of infection in Abstract and in the text when first mentioned.
  2. Figure 1 legend. “Bacteria (1:200 dilution) were cultured …” Not clear. 1:200 dilution from what concentration and what growth phase of bacteria??
  3. Line 191 and Table 3 “cathepsin G, proteinase-3, cathepsin B, and MMP-9” Need concentration and MOI ratio??
  4. Figure 2 legend: Section 2.2 shows no biofilm at high concentration of neutrophil, only lower concentration induce biofilm. Have authors done the titration to find out till what concentration of Proteinase-3 no biofilm detected?
  5. Line 291 and line 340 and Supplementary Figure 4 legend. “MOIs 0.001, 0.01” Not clear?? “neutrophils : bacteria” or “bacteria : neutrophils”??
  6. Line 318: Please spell PMA when first mentioned.
  7. Figure 9 legend is confusion, especially (A), (B), (C). P-3 in Figure 9A and 9B looks slightly different in mean and standard deviation.
  8. Line 484. “one of them the presence of neutrophils” should be “one of them is the presence of neutrophils”.
  9. Section 4.4. Line 517-519. How were the concentration determined?? Have you authors done the titration?

Author Response

Thank you, we made changes to the manuscript to incorporate the reviewer's comments.

  1. We thank the reviewer's suggestion, and we changed the title as recommended. We also made the solicited changes.
  2. Changed

  3. It was from a bacterial culture incubated overnight and diluted 1:200 with fresh TSB medium, and we changed figure 1 legend.
  4. The concentration of each protease was added in the foot of the table; please see the foot of the table. The table shows the total number of isolates that induced biofilm with the tested protease at any concentration tested. In other words, there were isolates that induced biofilm only at low concentrations of proteases, others induced biofilm up to intermediate concentrations, and others did so at all concentrations tested.

    Commercially purified proteases were used in this experiment, not neutrophil cells; therefore, there is no MOI ratio.
  5. In section 2.2, the induction of biofilm formation was done from a protein extract of neutrophil cells, for which we do not know the concentration of proteases in general, much less proteinase-3. Figure 2 shows the kinetics of biofilm formation at different concentrations in two isolates that gave biofilm formation in all tested concentrations of proteinase-3, and these were the chosen ones. However, as mentioned above (question 3), there were isolates that only induced biofilm at the first concentration of proteinase-3, indicating that the behavior is varied in all isolates, however in the two isolates used it was no biofilm formation.

  6. Throughout the text, it was added that the MOI is neutrophils: bacteria, 1:0.001, 1:0.01 rate, etc., to avoid confusion in both the text and figure captions.

  7. Changed.

  8. The figure 9 legend was modified to avoid confusion. In Figure 9A and 9B, P-3 is slightly different because they are separate trials.

  9. Changed.

  10. The concentration of commercial proteases was first determined by information from published works (not related to the article) where they suggest the use of low concentrations for each corresponding protease; then, based on this information, we had to adjust or titrate the concentration using a single isolate with the different low concentrations of the proteases until finding the concentration ranges used in this work, in addition, the other was the commercial presentation of the proteases that comes from very low concentration. As mentioned above, these concentration ranges tested at 53 isolates gave varied results that for those cases where biofilm was induced at all concentrations, titration was no longer performed.

Reviewer 2 Report

     This is a well made work of scientific interest, that explores the effects (defensive et al.) produced when neutrophils and Staphylococcus epidermis bacteria interact each other, with emphasis on the potential role of the former on the induction of biofilm formation, particularly through neutrophil proteinases, and on the generation of NETs. Comparisons were made between commensal (skin) and clinical isolated cell strains (the latter, mostly isolated from healthy conjunctiva, healthy skin, ocular infection and prosthetic joint infection), without or added neutrophils and derived proteinases (proteinase-3, cathepsin G, cathepsin B and MMP-9, as well as trypsin as control) or protease inhibitors (PMSF, pancreatic trypsin inhibitor, protease inhibitor cocktail), or DNAse, at different ratios and conditions, showing significant biofilm induction by the proteinases, particularly by proteinase-3. Kinetic studies on biofilm formation were also performed in such conditions, as well as previous genotyping and phenotyping characterization of the distinct isolated bacterial strains, focused on presence or absence of several gens encoding proteins related to biofilms and to the occurrence of these. Also, among other, in vivo experiments were done on biofilm formation, in a mouse with an infected catheter inserted in the back for seven days, confirming the higher number of viable bacteria recovered from the catheter at the end. Overall, a quite complete sets of experiments.

     Although the series of experiments performed can be followed reasonably well along the typescript, authors should include additional information on differents concepts (and implications) treated along the work which are quite specialized in the microbiology and immunology fields, to facilitate ample understanding and audience: i.e. about biofilms and NETs, roles and evaluation. Perhaps this work would have a better fit in more specialized journals of such fields, but given that the choice has been for IJMS, a journal dealing with fundamental problems on molecular aspects of broad interest in biology, chemistry and medicine, fitting with these ones should be facilitated.

-On this regard, and besides such concept clarification (i.e. on biofilm and NETs), the inclusion of an abbreviations section would also be helpful, to include terms like NETs, MOIs, PMA, PNAG/PIA, Aap, SepA, AtlE, CFU, ALP ...etc.

-Regarding the neutrophil proteases evaluated in the work, proteinase-3, cathepsin G, cathepsin B and MMP-9, it would be useful to briefly indicate their differential properties, like to which distinct clans/families/specificities are classified presently, and which kind of inhibitors should be used for them, and why certain ones have been selected for the present work (including the cocktail of inhibitors). At this respect authors should take into account that its is known that the selected PMSF inhibitor not only affects serine proteases but also cysteine proteases, and that nowadays is not advised its use for in cellulo o in vivo studies given its high toxicity (other related, as AEBSF, are better choices), low stability and solubility. Also, it should be indicated which is the particular commercial reference and supplier for such proteases and inhibitors; the particular case of "trypsin and trypsin inhibitor from bovine pancreas, from Sigma-Aldrich" (poorly cited, given the several variants available commercially), mentioned in lines 547-548, or the nothing or little said about the commercial or academic origin of the four main neutrophil proteinases, illustrate the need of better information about, to facilitate reproduction of the experiments, if wished. The same applies to the cocktail of proteinases, not referenced. By the way, the asseveration made at line 534 that proteases are classified into cysteine, serine and metalloprotease, a justification for the selection such three types of proteases for the present studies, is not fully true, giving that aspartic, threonine and other proteases are also considered distinct catalytic types. So the way to explain and/or justify such selection should be written in a different way.

-The last four lines of the Abstract (lines 47-50) are the same as the Conclusions (597-600). Besides of being a bit unusual practice (although probably acceptable), it seems that such procedure forgets the inclusion in the last section of certain conclusions collected in the first part of the Abstract, including the differential roles deduced for the other three neutrophil proteinases besides proteinase-3. These are points that should be improved.

-Authors tend to mix distinct mass or concentration units in the same sentences or paragraphs, in stead of homogenizing them: this is clearly seen in section 4.4, lines 517-518, in which mixing is made between μg/mL and ng/mL for the different proteases used: " ... 0.0001, 0.001, 0.01, or 0.1 μg/mL of cathepsin G; 0.25, 2.5 or 25 ng/mL of proteinase-3; 0.0001, 0.001, 0.01, or 0.1 μg/mL of cathepsin B and 0.01, 0.1, 1, or 10 ng/mL of metalloproteinase-9 (MMP-9) ...".   Why not to use only μg/mL or ng/mL?   (like ng: 0.1, 1.0, 10 or 100 ng/mL of cathepsin G; and 0.25, 2.5 or 25 ng/ml of proteinase-3 ...). Such a lack of homogeneity also appears in other parts of the typescript.

Author Response

We thank the reviewer's comments. We made modifications to the text according to your valuable suggestions.

  1. Information about biofilm and NETs was added. However, in the case of NETs, no more information was included because NETs were not the mechanism of biofilm induction by S. epidermidis, as is the case of Pseudomonas aeruginosa.
  2. An abbreviations section was included in the initial pages of the manuscript.

  3. In section 4.4, the classification of the proteases used is mentioned. The justification for selecting the proteases was also modified to read as follows: “Since activated neutrophils release a wide variety of proteases, their specific effect is not easy to study. Based on the work of Rohde et al. (2005) [20] that they used cathepsin G, elastase, trypsin, and based on the extracellular proteolytic system of S. epidermidis that expresses a cysteine protease (EcpA), a serine protease (Esp) and a metalloprotease (SepA) we used neutrophil proteases with these characteristics. Therefore, we selected for this work two representative serine proteases (cathepsin G and proteinase-3), a cysteine protease (cathepsin B), and a metalloprotease (metalloprotease-9, MMP-9), and trypsin as control.”

    Regarding the inhibitors, PMSF was used only to inhibit proteinase-3 in vitro in the presence of the bacteria. Concerning its toxigenic capacity, we observed no inhibition of bacterial growth (regardless of whether it forms a biofilm or not), which did not interfere with our results. It makes no sense to mention inhibitors for the other proteases since biofilm formation kinetics experiments were not performed with cathepsin G, cathepsin B, and MM-9 in the presence of their inhibitors.

    Neutrophil proteases were purchased commercially from Gibco, Thermo Fisher Scientific, MA, USA, trypsin, a trypsin inhibitor from bovine pancreas, PMSF, and the cocktail of proteinases from Sigma-Aldrich, Merck, State of Mexico, Mexico.

  4. The conclusion of the work was changed, and other points of the other proteases were mentioned.

  5. Protease units were homogenized to ng/mL.

Reviewer 3 Report

The neutrophil proteases Cathepsin G, cathepsin B, proteinase-2 3 and metalloproteinase-9 induce biofilm formation in non-bio-3 film-forming Staphylococcus epidermidis isolates by Itzia S. Gómez-Alonso et al.

This is a very nice paper with a high impact message describing the role of host factors in biofilm formation. The scientific set-up is robust, and the data are solid. Because of the complexity, it would be helpful to include a graphic summary of the research, depicting working mechanisms and interactions between the bacteria, biofilm formation, and neutrophils' role. Include NETs and eDNA and the stimuli and inhibitors.

Minor comments:

Explain the MOI and why this annotation is chosen above molarity or numbers of concentration series.

Explain the differences between NETs and the eDNA (of neutrophil or bacterial origin). This is sometimes confusing.

Describe the full name of PMA and the working mechanism.

Section 4.6: where were the bacteria injected?

Author Response

We appreciate your comments, and we modified the manuscrpt as suggested.

  1. The MOI was used to indicate the multiplicity of infection in a neutrophil:bacteria ratio, i.e., a ratio of 1:0.001, etc. In addition, MOI was used to fine-tune the ratio of neutrophil cells to bacterial cells. We included this information in the text.
  2. This part was modified in the text to understand that the extracellular DNA can come from the NETs or the broken bacteria.

  3. The PMA was described in the text.

  4. Bacteria were injected into the catheter, which was first introduced under the skin of the mouse's back; this information was added to avoid confusion.

Round 2

Reviewer 2 Report

     Authors made a substantial effort in revising and polishing the manuscript, satisfying most of the questions and requests raised.  Now it is of easier reading and understanding, and has improved internal coherence.